# Drug Interactions between Androgen Receptor Axis-Targeted Therapies and Antithrombotic Therapies in Prostate Cancer: Delphi Consensus

**DOI:** 10.3390/cancers16193336

**Published:** 2024-09-29

**Authors:** Kori Leblanc, Scott J. Edwards, George Dranitsaris, Darryl P. Leong, Marc Carrier, Shawn Malone, Ricardo A. Rendon, Alison M. Bond, Troy D. Sitland, Pawel Zalewski, Michelle Wang, Urban Emmenegger

**Affiliations:** 1Department of Pharmacy, University Health Network, Toronto, ON M5G 2C4, Canada; 2Leslie Dan Faculty of Pharmacy, University of Toronto, Toronto, ON M5S 3M2, Canada; 3Cancer Care Program, Eastern Health, St. John’s, NL A1B 3V6, Canada; scott.edwards@easternhealth.ca; 4School of Pharmacy, Memorial University of Newfoundland, St John’s, NL A1B 3V6, Canada; 5Department of Public Health, Falk College, Syracuse University, Syracuse, NY 13244, USA; george@augmentium.com; 6Hamilton Health Sciences, Population Health Research Institute, Hamilton, ON L8L 2X2, Canada; darryl.leong@phri.ca; 7Department of Medicine, McMaster University, Hamilton, ON L8S 4L8, Canada; 8The Ottawa Hospital Research Institute, Ottawa, ON K1Y 4E9, Canada; mcarrier@toh.ca; 9Department of Medicine, University of Ottawa, Ottawa, ON K1H 8L6, Canada; 10The Ottawa Hospital Cancer Centre, Ottawa, ON K1H 8L6, Canada; smalone@toh.ca; 11Department of Radiology, University of Ottawa, Ottawa, ON K1N 6N5, Canada; 12Queen Elizabeth II Health Sciences Centre, Halifax, NS B3H 3A7, Canada; rrendon@dal.ca; 13Department of Urology, Dalhousie University, Halifax, NS B3H 1Y6, Canada; sitlands@mac.com; 14Sunnybrook Health Sciences Centre, University of Toronto, Toronto, ON M4N 3M5, Canada; alison.bond@sunnybrook.ca; 15The Moncton Hospital, Moncton, NB E1C 4B7, Canada; 16Durham Regional Cancer Centre, Oshawa, ON L1G 2B9, Canada; pzalewski@lh.ca; 17Bayer Inc., Mississauga, ON L4W 5R6, Canada; michelle.wang@bayer.com; 18Odette Cancer Centre, Sunnybrook Health Sciences Centre, Toronto, ON M4N 3M5, Canada; uemmengg@sri.utoronto.ca; 19Department of Medicine, University of Toronto, Toronto, ON M5S 1A8, Canada

**Keywords:** prostate cancer, androgen receptor axis-target therapy, thrombosis, drug–drug interactions, anticoagulant

## Abstract

**Simple Summary:**

Prostate cancer is most commonly diagnosed in males after the age of 55 years. These patients are also at risk for cardiovascular disease and venous thromboembolism requiring antithrombotic therapy. Prostate cancer treatments, such as androgen receptor axis-targeted therapies (ARATs, i.e., abiraterone acetate, apalutamide, darolutamide, and enzalutamide), may interact with common antithrombotic medications like warfarin, clopidogrel, and the direct oral anticoagulants. However, the data detailing the clinical outcomes of patients treated with these combinations are limited. We undertook a comprehensive review of the literature and modified Delphi process to enable development of an evidence-based consensus document for the co-prescribing of ARATs with antithrombotic medications. Our assessments relied heavily on pharmacokinetic data and extrapolation from drug interaction studies of similarly metabolized drugs, highlighting the need for more research into the clinical impact of drug interactions in prostate cancer patients. Nonetheless, we provide a practical framework to support clinicians in day-to-day therapeutic decision making.

**Abstract:**

**Background/Objectives**: Abiraterone acetate, apalutamide, darolutamide, and enzalutamide, which make up the androgen receptor axis-targeted therapies (ARATs) drug class, are commonly used in the management of prostate cancer. Many patients on ARATs also receive oral antithrombotic therapy (i.e., anticoagulants or antiplatelets). The concomitant use of ARATs and antithrombotic therapies creates the potential for clinically relevant drug–drug interactions, but the literature regarding the actual consequences of these interactions, and guidance for co-prescribing, is limited. We assembled a multidisciplinary panel of experts and provided them with clinical information derived from a comprehensive literature review regarding the drug–drug interactions between ARATs and antithrombotic therapies. **Methods**: A three-stage modified electronic Delphi process was used to gather and consolidate opinions from the panel. Each stage consisted of up to three rounds of voting to achieve consensus on which ARAT/antithrombotic therapy drug pairs warrant attention, the possible clinical consequences of drug–drug interactions, and suggested actions for management. **Results**: The panel achieved consensus to avoid 11 ARAT/antithrombotic therapy drug pairs and modify therapy for eight pairs. Assessments relied heavily on pharmacokinetic data and extrapolation from drug–drug interaction studies of similarly metabolized drugs. **Conclusions**: This e-Delphi process highlights the need for further research into the clinical impact of ARAT/antithrombotic drug interactions. Nonetheless, the suggested actions aim to provide clinicians with a practical framework for therapeutic decision making.

## 1. Introduction

Prostate cancer is the most commonly diagnosed cancer in males in North America, with diagnosis occurring after the age of 55 years in more than 90% of cases [1,2,3]. Cardiovascular disease is also common in men of this age: 77% of men aged 60–79 years may have some form of cardiovascular disease [4,5]. Patients with malignancies, including prostate cancer, may be at even higher risk of cardiovascular disease and are also at an elevated risk for venous thromboembolism [6,7,8]. As such, patients with prostate cancer frequently receive antithrombotic therapy (ATT, i.e., anticoagulant or antiplatelet therapy) for cardiovascular disease and venous thromboembolism prevention and treatment.

Androgen receptor axis-targeted therapies (ARATs) represent a paradigm shift in the management of prostate cancer. ARATs, including abiraterone acetate, apalutamide, darolutamide, and enzalutamide, delay disease progression and improve overall survival in patients with advanced disease [9]. The metabolism of ARATs relies on the cytochrome P450 (CYP) system as well as transport proteins, such as P-glycoprotein (P-gp), the breast cancer resistance protein (BCRP), and the organic anion transporter polypeptide family member 1B1 (OATP1B1) [10,11,12]. These systems are also involved in the metabolism of certain oral ATTs [10,11,12]. In addition to being substrates, the ARATs also induce or inhibit various CYP enzymes and transporter proteins, increasing the likelihood of drug–drug interactions (DDIs) [10,11,12].

Guidance regarding the optimal management of these DDIs is limited. Clinicians managing patients with prostate cancer need to be aware of the risk of DDIs, and perhaps more importantly, need access to guidance on how to safely co-administer ARATs and ATTs. A recent publication reviewed pharmacokinetic (PK) information on these drug classes and predicted possible clinical events related to their PK interactions [13]. However it did not focus on management strategies for specific combinations of ARATs and ATTs.

Our goal was to develop an evidence-based consensus document for the co-prescribing of ARATs and ATTs in patients with prostate cancer, in the context of DDI risk. We undertook a modified electronic Delphi (e-Delphi) process, informed by a comprehensive review of the literature, to enable its development.

## 2. Materials and Methods

An e-Delphi process of scientific experts was conducted between September 2022 and August 2023. An e-Delphi process is a structured method used to gather and consolidate opinions from a panel of experts in order to reach a consensus by collecting feedback and refining opinions through multiple rounds of anonymously administered questionnaires [14]. Scientific experts for the e-Delphi panel were defined as qualified healthcare professionals with experience in prescribing and managing ARATs or ATTs and/or managing DDIs. The panel consisted of eleven experts: three pharmacists with respective expertise in hematology, oncology, and cardiology, three oncologists, two urologists, one cardiologist with a focus on cardio-oncology, one hematologist, and one non-voting methodologist. The methodologist developed the process, including preparing the questionnaires for each stage. Although the optimal size of a Delphi panel has not been established in the literature, prior studies have successfully developed consensus documents with at least 10 panelists [15,16].

The e-Delphi process was composed of three stages (Appendix A). Stage 1 consisted of selecting the most commonly encountered ATTs and ARATs; in other words, those deemed to be most relevant by panel members to be included in Stage 2. Stage 2 involved classifying the potential clinical consequences of DDIs resulting from the concomitant use of selected ARAT/ATT combinations as “Negligible”, “Minor”, “Moderate”, “Major”, or “Catastrophic”. To support this classification, a comprehensive literature review was undertaken. Stage 3 consisted of assigning suggested actions to manage ARAT/ATT drug pairs, for which consensus was achieved in the previous stage.

### 2.1. Questionnaire Development

Questionnaires specific to each stage of the e-Delphi process were developed using Microsoft Forms (https://www.microsoft.com/en-us/microsoft-365/online-surveys-polls-quizzes), pilot tested, and administered to panel members using a blinded web-based approach. Within each stage, the same questionnaires were administered up to three times, with group discussion between voting sessions, unless consensus was reached prior to three administrations.

### 2.2. Stage 1: Selection of ATTs for Inclusion

During a virtual meeting, the panel was presented with the four ARATs commonly used in prostate cancer: abiraterone acetate, apalutamide, darolutamide, and enzalutamide, as well as a list of available ATTs: the anticoagulants apixaban, dabigatran, edoxaban, rivaroxaban, heparin, dalteparin, enoxaparin, tinzaparin, fondaparinux, and warfarin, and the antiplatelet agents acetylsalicylic acid, clopidogrel, dipyridamole, prasugrel, ticagrelor, ticlopidine, and vorapaxar. The expert panel discussed whether to include each of these agents based on their perspective of the agents’ relevance to clinical practice in Canada. The panelists anonymously voted “Yes”, “No”, or “Not Sure” with respect to the inclusion of each ATT. The panel was also asked whether there were additional agents that should be considered which were not included in the list provided. After each round of voting, the results were presented such that each panel member could see the group outcome. Consensus was achieved when at least 80% of panelists voted “Yes” to include the ATT in Stage 2. ATTs achieving at least 80% consensus under the “No” category were not included. The remaining ATTs were put to the panel for further discussion before the subsequent round of voting was conducted.

### 2.3. Stage 2: Classification of Potential Consequences of DDIs

Data on the pharmacokinetics and DDI potential of the included ARATs and ATTs were identified through a comprehensive review of the English language literature. A search of PubMed was conducted in October 2022 to identify preclinical data, clinical trials, pharmacokinetic studies, cohort studies, case reports, and review articles published within the previous 20 years. A review of abstracts was conducted to exclude articles not including either information on the pharmacokinetics of target drugs/drug classes or use of target drug pairs/drugs classes of interest, articles in languages other than English, articles more than 20 years old, comments on primary research, dose prediction algorithms, drug level/efficacy correlations, evaluations of methodology, studies on the pharmacokinetics of alternative administration/formulations, pharmacogenomic studies, and pharmacokinetic studies in special populations, with the exception of cancer patients (Appendix A). Additional sources of information examined included American prescribing information [10], Canadian product monographs [11], European summaries of product characteristics [12], and Lexicomp, Micromedex, and CredibleMeds database search results for the drugs that progressed to Stage 2 [17,18,19], as well as the additional literature suggested by panelists based on their knowledge of the area and after review of the references retrieved through the literature search. Panel co-chairs (KL and SJE) summarized the key results of the literature search in a PowerPoint document, which was provided to the rest of the expert panel for review, along with copies of all source documents.

Following individual study of these documents, panel members undertook multiple rounds of voting on the potential consequences of DDIs, placing each pair into one of five categories, defined as follows:“Negligible”: available evidence, including pharmacokinetic and pharmacodynamic data, does not support an interaction with the concomitant use of these two agents;“Minor”: available evidence, including pharmacokinetic and pharmacodynamic data, suggests that there is, or may be, an interaction between these two agents but there is little to no evidence of clinical consequence, and the pharmacokinetics of these two agents suggest minimal theoretical clinical impact of the interaction;“Moderate”: available evidence, including pharmacokinetic and pharmacodynamic data, indicates that there is a real or strong theoretical interaction between these two agents that has resulted in or has the potential to result in either an increased incidence of adverse effects and/or reduced effectiveness of at least one of the agents;“Major”: available evidence, including pharmacokinetic and pharmacodynamic data, indicates that there is a clinically significant interaction between these two agents that is likely to cause harm (owing to adverse effects and/or reduced effectiveness of at least one of the agents) or require medical intervention to minimize or prevent serious adverse effects;“Catastrophic”: available evidence indicates that there is a clinically significant interaction between these two agents that has the potential to be life threatening; the combination of these two agents is generally considered contraindicated.

After each round of voting, the anonymized aggregate results were presented to the panel. Consensus was reached when at least 80% of panelists rated the DDI in either the upper (“Major” or “Catastrophic”), middle (“Moderate”), or lower categories (“Negligible” or “Minor”). Additional discussion was held if consensus was not reached. 

### 2.4. Stage 3: Assignment of Suggested Actions to Manage Drug Pairs

In Stage 3, the expert panel assigned suggested actions to manage possible DDIs for the drug pairings for which consensus was reached in Stage 2. The options for suggested actions consisted of “Avoid this pair”, which indicates that clinicians should consider an alternative ATT or ARAT, “Modify therapy”, which involves adjustment of the ATT dose to another approved dose or modification to treatment monitoring, or “No action needed”, meaning clinicians can continue therapy as planned. Additional discussion was held if at least 80% consensus was not reached on the actions assigned to individual pairs.

## 3. Results

### 3.1. ATTs Selected for Inclusion

Consensus was reached to include the oral anticoagulants apixaban, dabigatran, edoxaban, rivaroxaban, and warfarin, as well as the antiplatelet agent clopidogrel, in the next stages of the e-Delphi process. Consensus was reached to exclude heparin, dipyridamole, and ticlopidine. The most common reason cited for inclusion was the frequency of use observed in practice. No consensus was reached on the injectable anticoagulants dalteparin, enoxaparin, tinzaparin, and fondaparinux, or the oral antiplatelet agents acetylsalicylic acid, prasugrel, ticagrelor, and vorapaxar.

### 3.2. Comprehensive Literature Review

The search of the PubMed database identified 3910 articles, 3786 of which were removed after abstract review as they did not meet the pre-specified criteria for inclusion (Appendix B). The remaining 124 publications, along with an additional 39 publications identified by panel members, 14 American product information documents, 14 Canadian product monographs, 13 European summaries of product characteristics, and the results of searches of the Lexicomp, Micromedex, and CredibleMeds databases were made available to the panel (Figure 1). Panelists also received a document summarizing the key data from these references compiled by the panel co-chairs. The information from this document concerning the possible pharmacokinetic mechanisms of interaction for the drug pairs has been further summarized in Figure 2 and includes both in vivo and in vitro data.

### 3.3. Potential Consequences of DDIs

Overall, consensus on potential clinical consequence was achieved for 22 of the 24 drugs pairs, with 5 pairs being considered to have “Negligible”/“Minor” potential consequences, 10 having “Moderate” consequences, and 7 having “Major”/“Catastrophic” consequences (Figure 3). Consensus was reached on the potential consequences of DDIs for 9 of the 24 drug pairs during the first round of voting, 8 pairs during the second round, and 6 pairs during the third round (Appendix C). However, for the abiraterone acetate/clopidogrel pairing, there was discordance between the consensus achieved during Stage 2 (potential consequence being “Minor”) and that achieved in Stage 3 (suggested action being “Modify Therapy”), leading the expert panel to repeat Stage 2, which resulted in no consensus being achieved. In fact, after Stage 3 was completed and reviewed among all of the panel members, there was concern that these results did not align as a drug pair with minor clinical consequence that should not require therapy modification. Thus, the consensus from the panel was to repeat Stage 2 and Stage 3, neither of which gave rise to consensus.

Among the four ARATs, the results of the panel voting indicate that apalutamide, followed by enzalutamide, possessed the greatest potential for clinically relevant DDIs with ATTs based on their pharmacokinetic properties in combination with those of the ATTs. Among the direct-acting oral anticoagulants (DOACs), it was the consensus of the panel that apixaban and rivaroxaban possessed the greatest potential for clinically relevant DDIs with the ARATs (Figure 3).

### 3.4. Suggested Actions to Manage Drug Pairs

Overall, consensus on suggested actions was achieved for 22 of the 24 drug pairs (Figure 3); 18 pairs achieved consensus in the first round of voting, an additional 3 pairs in the second round, and the final pair in the third round (Appendix D). The panel suggested avoiding the concomitant use of 11 ARAT/ATT pairs and modifying dosing and/or monitoring for 8 pairs, with no action required for 3 pairs. It was the consensus of the panel that, as it related to ATT, pairs including apixaban, or rivaroxaban were those most often suggested to avoid. Pairs involving clopidogrel were the least likely to achieve consensus on the management of possible DDIs (Figure 3).

## 4. Discussion

Patients with prostate cancer receiving ARAT therapy often have cardiovascular and/or thromboembolic comorbidities requiring treatment with ATTs. However, the biotransformation of ARATs and ATTs share common pathways (Figure 2) and, as such, patients may experience clinically relevant DDIs resulting in decreased drug effectiveness or increased adverse effects. Our e-Delphi process reiterated the need for caution with regard to the combined use of certain ARATs and ATTs that has been previously raised by other investigators [13,27,28], and expanded on this by providing practical suggestions for actions to take for specific ARAT/ATT drug pairs.

### 4.1. Results of e-Delphi Process

The results of the e-Delphi process highlight that ARAT-ATT pairs involving apalutamide, apixaban, and rivaroxaban were those most often voted to avoid. Apalutamide strongly induces CYP enzymes involved in the metabolism of warfarin and certain DOACs and weakly induces transport proteins such as P-gp [10,11,12]. When used in conjunction with ATTs, this, in theory, may lead to decreased efficacy of oral anticoagulants, or may increase the activation of clopidogrel, increasing the risk of bleeding. While there is clinical evidence of apalutamide decreasing the international normalized ratio (INR) when combined with warfarin [10,11,12], it is important to note that there is little clinical outcome data that apalutamide decreases the efficacy of oral anticoagulants. In the case of warfarin, INR-guided dose-adjustment can mitigate the drug interaction. The inference of possibly harmful apalutamide/DOAC DDIs is mostly based on pharmacokinetic data and evidence of the clinical impact of the combination of DOACs with similar inducers of CYP3A4 and P-gp as apalutamide. The theoretical harmful result of reduced antithrombotic effect in a patient at significant risk of stroke or extension of a venous thromboembolism likely influenced our panel to err on the side of caution with respect to suggested actions to manage the potential interaction.

In cases where initiation of an ARAT is needed in a patient already stabilized on dabigatran, edoxaban, or clopidogrel, the consensus reached by the panel was that darolutamide would have the lowest potential for a clinically relevant DDI. Indeed, there is a pharmacokinetic study specifically demonstrating that there is no clinically relevant change to the pharmacokinetics of dabigatran, a P-gp substrate, in healthy volunteers despite darolutamide being listed as an in vitro P-gp inhibitor [22].

Conversely, in a patient already stabilized on an ARAT, the panel consensus was that, from a DDI risk perspective, dabigatran, edoxaban, or warfarin would be preferred if anticoagulation was required, as these anticoagulants appear to have a relatively lower risk for clinically relevant DDIs than apixaban and rivaroxaban. Dabigatran and edoxaban do not rely on CYP enzymes for metabolism to any clinically relevant degree [10,11,12]. Warfarin, although a substrate of several CYP enzymes, can be readily monitored and dose-adjusted according to the INR [10,11,12].

It is important to point out that the findings of this consensus document were intended to serve as a general guide for practicing clinicians. Several factors besides the drug pair can also influence the likelihood and clinical relevance of DDIs. Examples include the patient’s renal function, hepatic function, other concomitant interacting medications, indication, dose, and existing comorbidities. Both the clinical situation and individual patient risk factors need to be considered before a final management decision is made. Clinical situation-specific management may include adjusting the dose of an anticoagulant outside of current regulatory-approved doses using drug levels as a monitoring guide [29]. Alternatively, switching to a different ATT or ARAT might be reasonable as long as considerations such as indication, patient response, and drug access are taken into account. In cases of uncertainty, referral for specialist assessment may be helpful.

### 4.2. Strengths and Limitations of Current Work

Our e-Delphi process employed a systematic approach to utilizing the data from the literature search to inform the panel voting. A key advantage of the Delphi technique is that it helps overcome biases, groupthink, or dominance by individuals in a traditional group discussion setting. The iterative nature of the e-Delphi process allows experts to refine their views and consider alternative perspectives, leading to a more comprehensive and well-rounded outcome [14,30]. We utilized the pharmacokinetic data of each drug in the pair, evidence of the combination, and impact on pharmacokinetic and/or clinical outcomes where available. If not available, we utilized evidence of the ARAT combined with a drug with a similar pharmacokinetic profile to the ATT in question, or evidence of the ATT combined with a drug with a similar pharmacokinetic profile to the ARAT, and any available guidance on the specific pair provided by the regulatory drug product information, as well as the Lexicomp, Micromedex, and CredibleMeds databases.

Nevertheless, there are several limitations in our process that need to be acknowledged: (1) Most decisions relied upon extrapolation of consequences from published reports of ATTs with interacting drugs with similar pharmacokinetic profiles to the ARATs because little information regarding either the pharmacokinetic or clinical outcomes of the specific drug pair in combination was available. Further, where available, Phase 1 drug interaction studies were also used, which often involve healthy volunteers rather than typically elderly and often frail patients with prostate cancer. (2) The panel of experts was restricted to Canadian physicians and pharmacists, which may limit the generalizability of the recommendations to other countries. (3) Our results for the abiraterone/clopidogrel pair found discordance between the initial consensus result from Stage 2 versus that in Stage 3, as explained above in the Results section, which resulted in Stage 2 being repeated for that pair. However, despite this ad hoc modification to the methodology, the final result is that no consensus could be attained, which means that no guidance for the clinical management of this drug pair could be provided. (4) Finally, the e-Delphi process included only warfarin, DOACs, and clopidogrel, as other ATTs were either excluded via consensus or consensus was not reached on their inclusion/exclusion. Similarly, consensus was not reached on all the consequences and actions for selected drug pairs. Although this may be seen as limiting the clinical utility of our findings, it highlights important knowledge gaps, as well as the complexity of interpreting and managing DDIs.

## 5. Conclusions

In the context of the frequent co-occurrence of prostate cancer with cardiovascular disease and venous thromboembolism, clinicians require evidence-based guidance for the safe co-prescribing of concomitant ARATs and ATTs. Through an e-Delphi process, the consensus guidance from the multidisciplinary expert panel on the management of potential interactions suggested avoiding half of the pairs evaluated. However, these assessments relied heavily on pharmacokinetic data and extrapolation from DDI studies of similarly metabolized drugs, highlighting the need for further research into the clinical impact of DDIs in patients. Nonetheless, the practical suggested actions for the management of ARAT/ATT drug pairs should provide clinicians with a framework for therapeutic decision making.

## Figures and Tables

**Figure 1 cancers-16-03336-f001:**
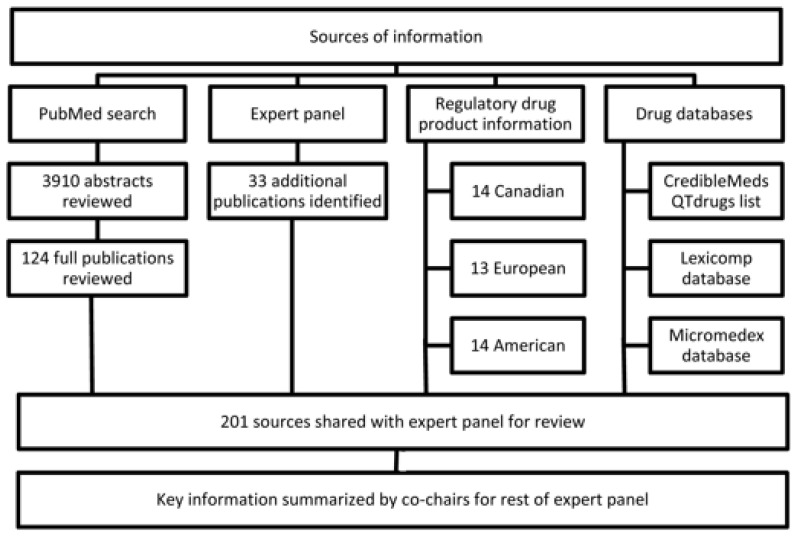
Consort diagram describing the literature review process. This consort diagram outlines the sources of information included in the literature review process, as well as the number of sources reviewed at each stage. Note: “Regulatory drug product information” refers to the Canadian product monographs, European summaries of product characteristics, and American product information documents that are approved as part of the marketing authorization of drug products in Canada, the European Union, and the United States, respectively. The CredibleMeds database monitors and analyzes data on drugs’ potential to elongate the QT interval or cause torsades de pointes, while Lexicomp and Micromedex offer interactive drug–drug interaction evaluation tools.

**Figure 2 cancers-16-03336-f002:**
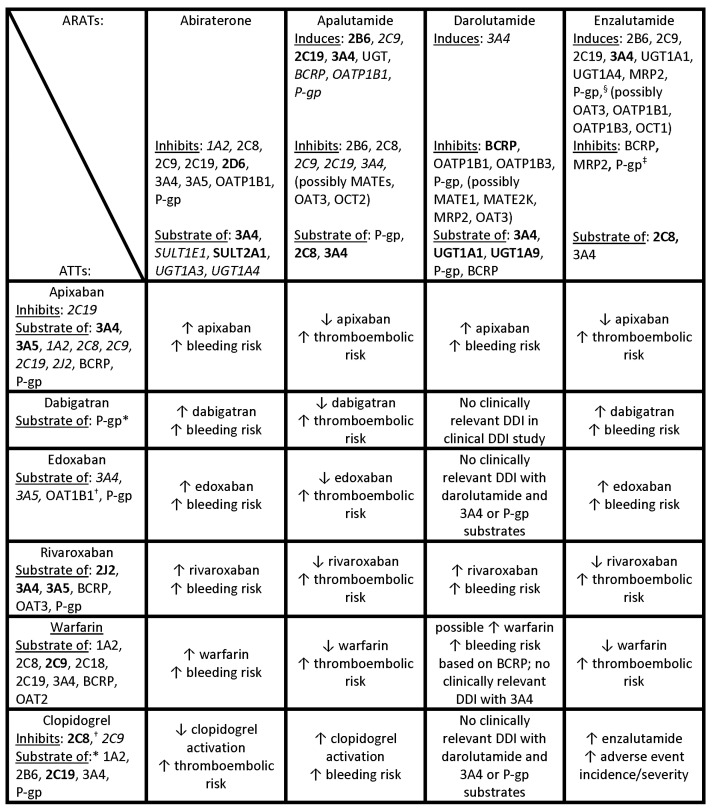
Pharmacokinetic mechanisms and theoretical DDIs for drug pairs [10,11,12,17,20,21,22,23,24,25,26]. This figure summarizes the results of the literature review process as it pertains to the published pharmacokinetic mechanisms of theoretical DDIs and includes both in vivo and in vitro data. **Bold text** indicates strong induction/inhibition or major substrate; *italic text* indicates weak induction/inhibition or minor substrate; Roman (plain) text indicates moderate or unspecified induction/inhibition/substrate. * Prodrug; ^†^ active metabolite; ^‡^ at higher concentrations; ^§^ at lower concentrations; ↑ increase; ↓ decrease ARAT, androgen receptor axis-targeted therapy; ATT, antithrombotic therapy; BCRP, breast cancer resistance protein; DDI, drug–drug interaction; MATE, multidrug and toxin extrusion; MRP, multidrug resistance-associated protein; OAT, organic anion transporter; OATP, organic anion transporter polypeptide; OCT, organic cation transporter, P-gp, P-glycoprotein; SULT, sulfotransferase; UGT, UDP-glucuronosyltransferase.

**Figure 3 cancers-16-03336-f003:**
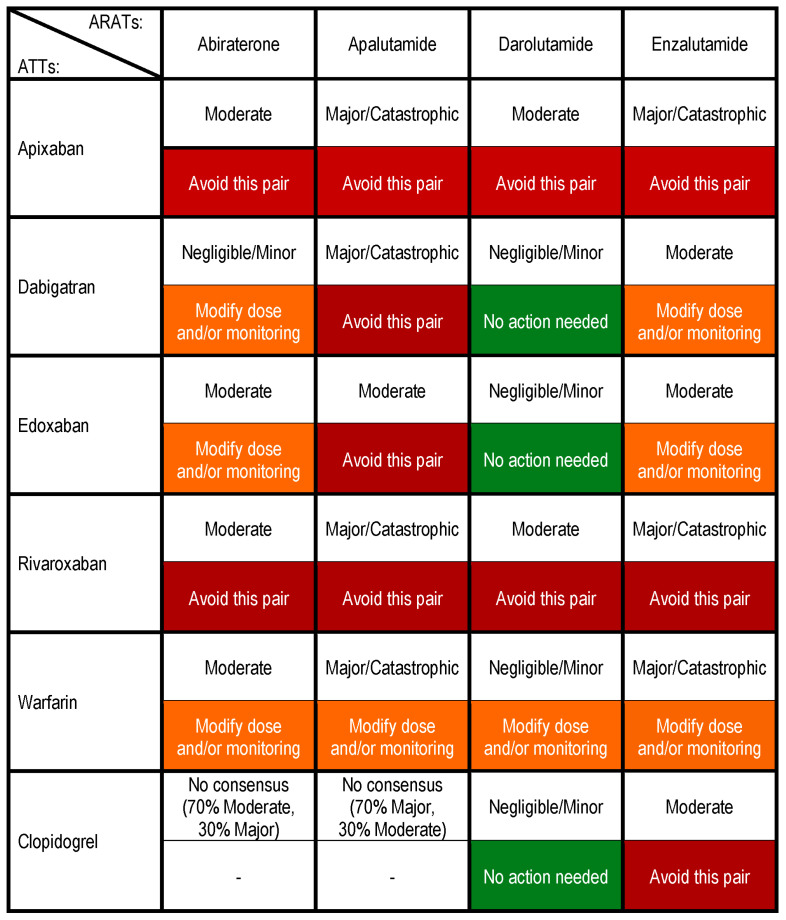
Potential consequences and suggested actions for ARAT/ATT drug pairs. This figure displays the results of the e-Delphi consensus process. The top row of each drug pair box (black font) indicates the consensus obtained with regard to the consequences of using the drug pair, while the bottom row (white font) indicates the consensus obtained for the suggested action to manage the drug pair; the results of the final round of voting are shown if no consensus was obtained. “Avoid this pair” (red boxes) indicates that one member of the drug pair should be changed, if possible. “Modify dose and/or monitoring” (orange boxes) indicates that the dose should be adjusted to another regulatory-approved dose and/or changes made to monitoring, as appropriate. “No action needed” (green boxes) indicates the drug pair can be used as planned with no additional actions taken. It should be noted that obtaining a specialist consult is an option in all cases. ARAT, androgen receptor axis-targeted therapy; ATT, antithrombotic therapy.

## Data Availability

The original contributions presented in the study are included in the article. Further inquiries can be directed to the corresponding author.

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
