# Peer review of "Drug Interactions between Androgen Receptor Axis-Targeted Therapies and Antithrombotic Therapies in Prostate Cancer: Delphi Consensus"

_cancers, 2024, doi:10.3390/cancers16193336_

Round 1

Reviewer 1 Report (New Reviewer)

Comments and Suggestions for Authors

Seems like authors have addressed most of the issues raised by the reviewers.   

Reviewer 2 Report (New Reviewer)

Comments and Suggestions for Authors

Authors are well summarized the current research on Drug Interactions Between Androgen Receptor Axis-Targeted Therapies and Antithrombotic Therapies in Prostate Cancer.

For readers convenience,  authors can add some figures or illustrations as summary of this review.

Comments on the Quality of English Language

Minor grammatical correction required throughout the review. 

This manuscript is a resubmission of an earlier submission. The following is a list of the peer review reports and author responses from that submission.

Round 1

Reviewer 1 Report

Comments and Suggestions for Authors

I would like to thank the authors for their answers to the comments. However, I do believe that we are not align on several points.

Author Response

Comment:  I would like to thank the authors for their answers to the comments. However, I do believe that we are not align on several points.

Author Reply:  It is difficulty to address this comment without specific areas to improve.  We provided a reply to each of the major comments in the first round of reviews, and note that there are no longer any sections of the manuscript that meet the "must be improved" category in this round of review.  It is our hope that, with our response to the Academic Editor's comments, we may align our work better with the perspective of Reviewer 1.

Reviewer 2 Report

Comments and Suggestions for Authors

The authors have made sufficient changes according to the suggestions

Author Response

Comments: The authors have made sufficient changes according to the suggestions

Author Reply:  Thank you.  We sincerely appreciate the opportunity to address the comments.

Reviewer 3 Report

Comments and Suggestions for Authors

In this article, LeBlanc et al employed a three-stage modified electronic Delphi process to achieve consensus on which androgen receptor axis-targeted therapy/antithrombotic therapy drug pairs warrant attention in the clinic. The panel of experts that participated in this study identified 11 drug pairs that need to be avoided, 8 pairs that needed monitoring or dose adjustment and 3 pairs that are safe to be used. The manuscript is well written; study strengths and limitations are clearly defined, and it can be used as a guide for more in-depth clinical research.

Minor comments

-Line 82: Please add the reference.

-Section 2.1. If possible, please upload the sample questionnaires that we used to survey the panel experts involved in this study.

Author Response

Comments: 

In this article, LeBlanc et al employed a three-stage modified electronic Delphi process to achieve consensus on which androgen receptor axis-targeted therapy/antithrombotic therapy drug pairs warrant attention in the clinic. The panel of experts that participated in this study identified 11 drug pairs that need to be avoided, 8 pairs that needed monitoring or dose adjustment and 3 pairs that are safe to be used. The manuscript is well written; study strengths and limitations are clearly defined, and it can be used as a guide for more in-depth clinical research.

Minor comments

-Line 82: Please add the reference.

Author Response:  Thank you for your review of our paper. The reference has been added to line 82.

-Section 2.1. If possible, please upload the sample questionnaires that we used to survey the panel experts involved in this study.

Author Response:  The questionnaires used in our panel survey (voting) were online forms as described in the manuscript and the voting was done in a virtual environment.  However we have converted those files to print and have attached to the manuscript as a supplementary appendix.  Thank you for the suggestion.

Reviewer 4 Report

Comments and Suggestions for Authors The study is not evidence based and the authors did not address fully the critiques of the two reviewers. My evaluation of the manuscript:  Is not fit to be published in Cancers.

Author Response

Comment 1:

The study is not evidence based and the authors did not address fully the critiques of the two reviewers

Author Reply:  We respectfully disagree that our work is not evidence based.  It would be helpful to understand the reviewer's perspective as to what specific components of our methodology do not, in the reviewer's view, meet their definition of 'evidence based'.   Further, reviewer 2 accepted our reply and revisions, deeming them acceptable.  We have replied to the Academic Editor's review expanding on the concerns of Reviewer 1.

We agree that there is a component of the Delphi method, that relies on the opinion of experts to achieve consensus on an outcome - in our case, an opinion on whether an interaction between two medications exists and has the potential to cause clinical consequences, and what, as clinicians treating patients with these combinations of medications would recommend when faced with a patient requiring both classes of medication.  The information provided to the experts involved was indeed evidence based - and the experts involved accepted the responsibility to ensure that they utilized the 201 reference papers to inform their voting.

Comment 2: 

My evaluation of the manuscript:  Is not fit to be published in Cancers.

Author reply:  We would be grateful for specific comments to where we could improve the manuscript.  We are disappointed in the comment but respect the review process and thank you for your time.